# The Optimal Choice of Trap Type for the Recently Spreading Jewel Beetle Pests *Lamprodila festiva* and *Agrilus sinuatus* (Coleoptera, Buprestidae)

**DOI:** 10.3390/insects14120961

**Published:** 2023-12-18

**Authors:** Eszter Matula, Gábor Bozsik, József Muskovits, Csenge Ruszák, Laura Jávorszky, Jochem Bonte, Márton Paulin, József Vuts, József Fail, Ágoston Tóth, Ádám Egri, Miklós Tóth, Zoltán Imrei

**Affiliations:** 1Plant Protection Institute, Centre for Agricultural Research, HUN-REN, Herman O. Street 15, H-1022 Budapest, Hungary; matula.eszter@atk.hun-ren.hu (E.M.); gabor.bozsik@atk.hun-ren.hu (G.B.); miklos.toth@atk.hun-ren.hu (M.T.); 2Department of Entomology, Institute of Plant Protection, Hungarian University of Agriculture and Life Sciences, 44 Ménesi Street, H-1118 Budapest, Hungary; 3Independent Researcher, H-1119 Budapest, Hungary; 4Biocont Magyarország Kft., 1 Trafó Street, H-6000 Kecskemét, Hungary; 5Plant Sciences, Flanders Research Institute for Agriculture, Fisheries and Food (ILVO), 9820 Merelbeke, Belgium; 6Department of Forest Protection, Forest Research Institute, University of Sopron, H-3232 Mátrafüred, Hungary; 7Department of Biointeractions and Crop Protection, Rothamsted Research, Harpenden AL52JQ, UK; jozsef.vuts@rothamsted.ac.uk; 8Institute of Aquatic Ecology, Centre for Ecological Research, HUN-REN, Karolina Road 29, H-1113 Budapest, Hungary

**Keywords:** jewel beetle, *Agrilus*, *Lamprodila*, *Ovalisia*, sticky trap, invasive pest

## Abstract

**Simple Summary:**

Two jewel beetle species native to Europe, the cypress jewel beetle, *Lamprodila* (*Palmar*, *Ovalisia*) *festiva* L. (Buprestidae, Coleoptera), and the sinuate pear tree borer, *Agrilus sinuatus* Olivier (Buprestidae, Coleoptera), are key pests of ornamental thuja and junipers and of orchard and ornamental rosaceous trees, respectively. Although chemical control measures are available, due to the beetles’ small size, agility, and cryptic lifestyle at the larval stage, efficient tools for their detection and monitoring are missing. Consequently, by the time emerging jewel beetle adults are noticed, the trees are typically significantly damaged. Thus, the aim of this study was to initiate the development of monitoring traps. Transparent, light green, and purple sticky sheets and multifunnel traps were compared in field experiments in Hungary. Light green and transparent sticky traps caught more *L. festiva* and *A. sinuatus* jewel beetles than non-sticky multifunnel traps, regardless of the larger size of the colored surface of the funnel traps. Although light green sticky sheets turned out to be optimal for both species, using transparent sheets can reduce catches of non-target insects. The key to the effectiveness of sticky traps may lie in the behavioral responses of the beetles to the optical features of the traps.

**Abstract:**

BACKGROUND: Two jewel beetle species native to Europe, the cypress jewel beetle, *Lamprodila* (*Palmar*, *Ovalisia*) *festiva* L. (Buprestidae, Coleoptera), and the sinuate pear tree borer, *Agrilus sinuatus* Olivier (Buprestidae, Coleoptera), are key pests of ornamental thuja and junipers and of orchard and ornamental rosaceous trees, respectively. Although chemical control measures are available, due to the beetles’ small size, agility, and cryptic lifestyle at the larval stage, efficient tools for their detection and monitoring are missing. Consequently, by the time emerging jewel beetle adults are noticed, the trees are typically significantly damaged. METHODS: Thus, the aim of this study was to initiate the development of monitoring traps. Transparent, light green, and purple sticky sheets and multifunnel traps were compared in field experiments in Hungary. RESULTS: Light green and transparent sticky traps caught more *L. festiva* and *A. sinuatus* jewel beetles than non-sticky multifunnel traps, regardless of the larger size of the colored surface of the funnel traps. CONCLUSIONS: Although light green sticky sheets turned out to be optimal for both species, using transparent sheets can reduce catches of non-target insects. The key to the effectiveness of sticky traps, despite their reduced suitability for quantitative comparisons, may lie in the behavioral responses of the beetles to the optical features of the traps.

## 1. Introduction

Both the cypress jewel beetle, *Lamprodila* (*Palmar*, *Ovalisia*) *festiva* L. (Buprestidae, Coleoptera) [1,2,3] and the sinuate pear tree borer (also called hawthorn jewel beetle), *Agrilus sinuatus* Olivier (Buprestidae, Coleoptera) [4], are native to Europe and also occur in North Africa to the south and in the Caucasus to the east.

*L. festiva* used to be considered economically indifferent, with rare records of its presence. Before 2010, *L. festiva* was known to attack only wild trees from the Cupressaceae family and had a distribution limited to the surroundings of the Mediterranean Sea (including North Africa) and parts of France [5]. It was reported rarely and only in low population densities in other European regions and was a protected red-list species in many European countries [1]. Probably warmer, drier periods in the last two decades led to its increased abundance and spread on a continental level, reaching as far as the Black Sea coast of the Caucasus [2,6] in the east and the Netherlands, Luxemburg, and Germany in the north [2,7]. *L. festiva* subsequently gained pest status on ornamental plants, typically *Thuja* and *Juniperus* species planted at suboptimal locations [8,9] and relict forests [6]. The spread of *L. festiva* has led to a halt on growing white-cedar (*Thuja occidentalis* L.), common juniper (*Juniperus communis* L.), and other related species in nursery plantations in Hungary (Z. Imrei, unpublished).

The life cycle of *L. festiva* is 1–4 years, depending on temperature and humidity. According to existing literature, the adults are active in May–July [10,11], occasionally until August [12]. After mating, the female oviposits in the small cracks of the host plant’s bark on trunks or branches, predominantly on the sun-exposed sites [12]. The ovipositor is soft and membranous, with two apical palpators, which are used to check for cracks or crevices in the bark [10,11]. The larvae mine under the bark and feed on the cambium, then the sap wood. The tunnels are flat and about three times wider than the thickest part of the larva itself. Larvae develop in the lower parts of branches or trunks, where they dig sinuous sub-cortical galleries that affect the phloem tissue, deteriorate the xylem, and decrease the mechanical integrity of the trunk and branches [12]. The larvae can severe a large amount of xylem vessels, so for example, ten larvae can kill a 7 m tall white cedar [10,11]. The mature larva digs deeper into the sapwood to make the pupal chamber. The pupa is oriented with the head toward the tree exterior. The emergency holes are “D” or oval-shaped and 2–3 mm in the longer diameter.

*A. sinuatus* was mentioned as an occasional pest in the 19th century (1875), but it was not until the mid-20th century (1949) that it caused severe damage to fruit farms and nurseries in Europe [13]. Subsequently, in the era of broad-spectrum insecticides, *A. sinuatus* became rare, while from 1992 in Germany, 1996 in the Netherlands, and 1997 in Belgium, large populations caused occasional damage to orchards and ornamental trees [14]. Its population growth could be explained by more favorable climatic conditions, such as several successive hot summers, as the species is thermophilic.

*A. sinuatus* is currently considered a major pest of pear in Belgium [4] and neighboring countries, as the larvae can cause primary damage by killing the host trees by making a typically downward-running sinuous gallery just below the bark [13,14]. *A. sinuatus* is harmful to both young plantations and older trees, both in nurseries and orchards [4]. The bark of damaged trees becomes necrotic and dies off, creating cracks that provide entry points for various secondary pathogens. *A. sinuatus* is known to develop in wild pear (*Pyrus pyraster* L.) and is a pest of pear (*Pyrus communis* L.) [13,14], ornamental hawthorn (*Crataegus* spp.) [13,15,16], rowan (*Sorbus* spp.) [4], medlar (*Mespilus germanica* L.) [13,14], and quince (*Cydonia oblonga* Mill.) [13,14]. Neither apple (*Malus*), nor plum (*Prunus*), or any stone fruit trees, are host plants [4,13,14]. Certain pear varieties are known to be particularly sensitive, including ‘William’s Good Christian’, ‘Doyenné du Comice’, ‘Conférence’, ‘Saint Rémy’, ‘Légipont’, and ‘Louise Bonne d’Avranches’; the ‘Carisii’ graft intermediate is very sensitive, while *P. calleryana* Decne. Var. ‘Chanticleer’ seems to be resistant [13].

From 2019, an immense growth of an *A. sinuatus* population was recorded in a 4 ha quince orchard at Hajdúnánás, Hungary, resulting in significant damage that led to 40–50% tree mortality and the general weakening of the whole *quince* orchard (Cs. Ruszák and Á. Tóth personal observations), similarly to the damage reported in pears from Belgium [14]. In 2021, the trees were eventually cut down because of a lack of profitability.

Females lay 30–40 eggs on the sunny side of young tree trunks or thicker branches [4,13]. Under the bark of 3–6 cm-long hawthorn branches, they bore wavy tunnels that are clearly apparent when the bark is peeled off. A tunnel, which can be 50–60 cm in length, usually progresses downward, only rarely upward [4,14]. At the end of the second autumn, the larvae develop fully and prepare a pupal chamber, usually 2–3 cm below the wood surface. Before pupating, the larvae chew in the wood through an exit hole, which they fill with remnants of chewing without damaging the bark. The adults hatch from the end of May to the beginning of June and can be found on their host plants until the end of July [4].

As efficient tools for the detection or monitoring of *L. festiva* and *A. sinuatus* are lacking to date [13,14,17,18], our aim was to develop traps for the detection and monitoring of the two jewel beetle species. A key requirement was that the traps be easy to handle and cost-effective, which would make them economically more viable. Light green and purple colors were chosen as visual cues for the field trials because the light green color was demonstrated to be a strong general attractant for a wide range of *Agrilus* spp. in Europe [19] and in North America [20,21], whereas the purple color was found to attract *Agrilus biguttatus* and Fabricius jewel beetles in oak woodlands in the U.K. [22] and the emerald ash borer *Agrilus planipennis* Fairmaire in North America [20,21]. For *L. festiva* in 2019 and 2020, sticky and non-sticky traps with or without the light green or purple color were compared, whereas for *A. sinuatus* in 2021, sticky and non-sticky traps with or without the light green color were compared to study the relative importance of visual cues and trap surface characteristics.

## 2. Materials and Methods

### 2.1. Trap Type

*Sticky cloak trap types (commercial CSALOMON^®^ transparent PAL and light green PALz):* Sticky traps were prepared using 23 × 36 cm plastic sheets left either transparent (PAL; Figure 1A) or painted light green (PALz; Figure 1B; produced by Plant Protection Institute, Centre for Agricultural Research, HUN-REN, Budapest, Hungary, www.csalomontraps.com, accessed on 15 December 2023). Plastic sheets were covered on one side with sticky glue (Tangletrap Insect Trap Coating, The Tanglefoot Company, Grand Rapids, MI).

*Multifunnel trap types (transparent MULT*, *light green MULTz*, *and purple MULTp):* The upper funnels of the commercial CSALOMON^®^ VARb3 funnel trap (produced by Plant Protection Institute, Centre for Agricultural Research, HUN-REN, Budapest, Hungary, www.csalomontraps.com, accessed on 15 December 2023) were arranged in a vertical position on top of each other at approximately 15 cm spacing, using four identical upper funnels. The inside of the funnels was painted light green (MULTz; Figure 1C) or purple (MULTp). For the reflectance spectrum of the purple color, which is not discussed here, see Brown [23]. Unpainted funnels were used to prepare the transparent control traps (MULT; Figure 1D). The inside of the funnel parts was coated with Teflon^®^ (95% polytetrafluoroethylene-based spray; B’laster Corporation, Cleveland, OH, USA) to increase trapping efficiency by providing a slippery surface for beetles, which fall into the catch container at the bottom of the trap upon landing [24]. A piece of Vaportape^®^ insecticidal strip (Hercon Environmental Inc., Emigsville, PA, USA) was placed in the collection bucket. The weight of the MULTz, MULTp, and MULT trap designs is about 410 g.

### 2.2. Experimental Site and Setup

Experiments on *L. festiva* were conducted at Tahi Tree Nursery in Hungary (Tahi, Pest County, Central Hungary, GPS 47.7661127 N, 19.0684354 E). Experiment 1 (Exp. 1) was conducted between 12 June and 26 July 2019. Traps were set up in rows of *T. occidentalis* var. Smaragd trees at a nursery plantation in a randomized complete block design, with four replicates of sticky traps (PAL, PALz) and three repetitions of multifunnel traps (MULT, MULTz, MULTp). Experiment 2 (Exp. 2) was conducted between 4 June and 6 August 2020. The setup of traps was similar to Exp. 1, with 6 replicates for all trap types (PAL, PALz, MULT, MULTp, and MULTz). In Exp. 1 and 2, the corners of the PALz and PAL were fastened to the twigs of *T. occidentalis* trees, stretched and tight fit to the foliage (Figure 1A,B), at a height of about 180 cm, and multifunnel trap types were suspended by a piece of wire attached to a horizontal bamboo stick and fixed to the trees at 200 cm height. The height of trees in the experimental field was cc. 4 m. Traps were always placed on the sunlit sides of the trees.

Experiment 3 (Exp. 3) on *A. sinuatus* was conducted at Hajdúnánás in Hungary (Hajdú-Bihar County, Eastern Hungary, GPS 47.8438028 N, 21.4104173 E) between 23 June and 19 July 2021. Traps were set up in every second row of a quince plantation. Initially, traps were set up on 23 June in a randomized complete block design on sunlit sides of the trees, with four replicates of each trap type (PAL, PALz, MULT, and MULTz). Following the first inspection on 30 June 2021, 10 additional replicates of PALz and PAL traps were set up, evenly distributed between blocks, as only sticky traps caught any *Agrilus* spp. at the first inspection.

In Exp. 3, PAL and PALz traps were bended in a “cloak-like” manner, with the sticky side facing outward, and both multifunnel and sticky traps were hung from a branch of a quince tree at about 2 m height. For all experiments, traps were spaced 5 m apart within a block, and blocks were located 10 m apart. Traps were inspected once a week; any jewel beetles (Coleoptera, Buprestidae) captured were removed and identified as species in the lab. Non-jewel beetle catches, mostly flies (Diptera), were removed without identification.

### 2.3. Reflectance of the Traps

For the human eye, the difference between a sticky and a non-sticky surface of a given type is manifested in the glittering caused by the irregular surface of the glue in the case of the sticky surface. To optically characterize this glitter, we performed spectral measurements on sticky (PAL) and non-sticky (MULT) transparent and light green-painted PALz and MULTz surfaces, respectively. All four surfaces were fixed on a black, vertically aligned piece of cardboard functioning as a background (Figure 2). The illumination was provided by the skylight without direct sunlight. An Ocean Optics STS-VIS spectrometer equipped with a P400-010-UV-VIS fiber (Ocean Optics, Largo, FL, USA) and a Spectralon^®^ white diffuse reflectance standard (Edmund Optics Inc., Barrington, IL, USA) was used for the measurements. Spectralon^®^ is a diffuse reflector possessing a constant 99% reflectance over a wide spectral range, including the region of insect vision. The field of view of the fiber was 30 degrees. The surface of the Spectralon^®^ was similarly vertical as the surfaces, and the optical axis of the fiber was horizontal during the measurements. The ratio of measured spectra on the Spectralon^®^ and on a given surface resulted in the reflectance spectrum. The distance between the fiber terminal and the reflecting surface was 5 cm. A total of 10 measurements were made on each surface (location: 47°28′43.586″ N, 19°1′51.99″ E, time: 28 July 2023, 12:20 UTC + 2 h).

### 2.4. Statistics

Statistical analysis was conducted in R version 4.3.0 [25], and figures were produced using the software packages “dplyr” (Version: 1.1.4.) [26] and “ggplot2” (Version: 3.4.4) [27]. As even transformed data did not meet the assumptions of parametric tests, the non-parametric Kruskal–Wallis test was used [28]. When the Kruskal–Wallis test indicated significant differences (*p* = 5%), pairwise comparisons by Wilcoxon test were conducted [29].

## 3. Results

### 3.1. Behavioral Field Tests

*L. festiva.* Altogether, 505 *L. festiva* specimens were caught during Exp. 1 (Figure 3A, Table A1). Significantly more beetles were captured by sticky PAL traps compared to non-sticky MULT, MULTp, and MULTz traps. More beetles were caught by sticky PALz traps than by non-sticky MULT or MULTp, and in absolute numbers, MULTz traps showed similar tendencies.

Altogether, 118 *L. festiva* specimens (73 males and 45 females) were caught during Exp. 2 (Figure 3B, Table A1), where significantly more *L. festiva* specimens were captured by sticky traps as compared to multifunnel traps, irrespective of color. No significant difference was found between PAL and PALz. Catches of males or females showed the same tendencies; the only exception was when comparing multifunnel traps; significantly more male beetles were caught in MULTz traps than in MULT traps (Figure 4A,B).

*Agrilus sinuatus.* In total, 263 *A. sinuatus* specimens were caught during Exp. 3 (Figure 5, Table A1), significantly more in sticky traps compared to multifunnel traps, and a trend of more catches in absolute numbers was observed when the light green visual stimulus was present in both sticky and non-sticky traps.

### 3.2. Reflectance of the Traps

Figure 6A shows the mean reflectance spectrum of the sticky and non-sticky transparent trap surfaces mounted on the black cardboard. The reflectance of the non-sticky transparent surface is very low in the 350–700 nm measurement range, which is not surprising because the background was black cardboard. On the other hand, the reflectance spectrum of the sticky transparent surface is very similar in shape but elevated compared to that of the non-sticky surface. The reflectance increase is shown in Figure 6C, which is the difference between the solid and dashed curves in Figure 6A. It is clear that the stickiness implies an approximately 4–5% increase in reflectance at all wavelengths, and there is a slight wavelength dependence showing a greater increase in reflectance towards the shorter wavelengths.

Practically the same results were obtained for the green-painted traps. Figure 6B shows the mean reflectance spectrum of the sticky and non-sticky green-painted trap surfaces. Similar to the case of the transparent surfaces, the presence of the sticky glittering glue resulted in an approximately 4% increase in reflectance (Figure 6D). This increase is more or less wavelength-independent; however, a slight increase towards the shorter wavelengths is also observable.

## 4. Discussion

In the present experiments, transparent PAL and light green PALz sticky traps were more efficient than non-sticky funnel traps for catching both *L. festiva* and *A. sinuatus* jewel beetles, despite the combined larger colored surface of the non-sticky trap designs. This suggests that sticky traps are more suitable for the monitoring of these two pest species of jewel beetles. According to the practice developed in our experiments, sticky sheets should be stretched and tightly fit to the foliage for *L. festiva*, whereas they should be bent in a “cloak-like” manner for *A. sinuatus*. Light green sticky sheets seem to be optimal for both species; however, non-target insect catches can be reduced using transparent sheets. In any case, these two methods could form the basis for monitoring *L. festiva* and *A. sinuatus*.

Despite the fact that non-sticky trap designs can make handling and identification of specimens far easier [19], and they are more suitable for quantitative comparisons than sticky designs [30], the results of the present study demonstrate that funnel traps are not suitable for catching these jewel beetles. The better performance of sticky sheets compared to funnel traps could be due to species-specific beetle behavior, such as their ability to avoid or escape from funnel systems or their lack of ability to recognize shiny sticky surfaces as potential threats. Further, the optical features of the sticky material may, in fact, attract these species. On the contrary, according to literature data and our own experience [17], multifunnel traps (light green MULTz and 12-funnel green Lindgren) are at least as effective as sticky light green PALz traps for trapping several oak-dependent *Agrilus* spp., including *A. obscuricollis* Kiesenwetter, *A. graminis* Laporte et Gory, *A. angustulus* (Illiger), *A. laticornis* (Illiger), *A. litura*, *A. olivicolor* Kiesenwetter, and the ash-dependent *A. convexicollis* Redtenbacher [17], and *A. planipennis* [31].

According to our spectral measurements, the sticky version of a given surface reflects 4–5% more light compared to its non-sticky counterpart. This increase is due to the irregular, lumpy surface of the glue, which reflects light in various directions. It means that a portion of the bright skylight easily gets reflected to the eye of the observer, even if the orientation of the surface itself would not allow it (Figure 5A,B). This increase in intensity results in a brighter surface for the observer. Consequently, the achromatic contrast between the background (usually the green foliage) and a trap surface is higher in the sticky case, which makes the surface more conspicuous.

The spread of *L. festiva* draws attention to the fact that biological invasions are one of the most critical problems today [32] in tree- and shrub-dominated ecosystems [33]. Even though bark beetles and borers comprise a relatively small proportion of invasive tree and shrub pests, their damage is far more significant. Extensive phytosanitary measures are needed to prevent their introduction into new regions [18]. In the case of many high-risk jewel beetle species, it is necessary to monitor them both where they are endemic and where they have been introduced to prevent unexpected and irreversible tree destruction, which may even affect entire continents, or simply to measure their effect on tree health [14,20]. There is an urgent need to prepare for these pests in Europe, as in the case of *A. planipennis*, where a potential invasion from European Russia is expected [34]. This will most likely lead to an environmental challenge, significantly changing urban areas, forests, and other agroecosystems where ash stands [35]. Although not quarantine organisms, there is a similar need for the development of suitable monitoring methods for *A. sinuatus* [13] and *L. festiva* [2] that support timely detection for efficient plant protection measures to reduce their damage to ornamental and fruit host trees.

Our visual observations of *L. festiva* indicated that the peak of flying activity occurs during sunny afternoon hours in the trapping area, resulting in a mean of 15 flying adults at a ten-minute visual inspection conducted walking through the approximately 20-m-long rows of *T. occidentalis* var. Smaragd plantation from the beginning of June to the end of July in 2019. This observation was made solely for supplementary information. Revealing the daily rhythm of the beetle was not the aim of the study. There were over four times more *L. festiva* captured in 2019 compared to 2020, a yearly decrease that is considered usual among insects. However, at the end of 2019 and in early 2020, several rows of *T. occidentalis* trees were chopped in the nursery as they lost their ornamental value, which reduced the occurrence of host trees significantly in the vicinity of our experimental field and could affect the absolute number of *L. festiva* specimens as well.

Although *Agrilus* spp., such as *A. sinuatus*, can only be controlled with chemical measures in their adult stage or as a young larva [14], efficient tools for its detection or monitoring, including recognition of establishment at newly invaded sites, are missing [13], similarly to *L. festiva* [18] and several other pest jewel beetles [17,18]. This is due to their small size, being swift flyers (not easy to observe), and because the larvae develop hidden under the bark of trees, resulting in their presence being invisible to unexperienced eyes [13,36]. Phytosanitary inspectors are faced with similar problems during visual inspections at high-risk sites. Due to such difficulties, jewel beetle infestations often go undetected during initial colonization, resulting in insidious damage [14,37]. This is a typical case for *A. sinuatus* in pears [14] and in quince at the Hungarian site of the present study. By the time the characteristic exit holes are recognized and the emerging adults are noticeable, the trees are usually already significantly damaged. However, based on the present results, basic methods are now available for the detection of both *L. festiva* and *A. sinuatus* to specify their exact period of emergence, to which treatments can be timed as soon as the first adults emerge before egg-laying. Also, the duration of swarming, i.e., the period during which intervention may be required, can be determined. Previously, Fassotte [14] reported a number of unsuccessful detection and monitoring trials with different trap types and stimuli. Once *A. sinuatus* is detected by the sticky PAL or PALz traps described in the present study, the first treatment should be carried out no earlier than one week after the first recorded emergence to pinpoint the time of egg-laying [4]. Given the long flight period that lasts for at least two months in Belgium and in the Netherlands, it was recommended to carry out repeated chemical treatments, at least four applications, during the entire flight period, the interval of which depends on the length of the insecticide activity of the products used.

## 5. Conclusions

The methods presented here are preliminary approaches for monitoring *L. festiva* and *A. sinuatus*, which can be further improved and optimized for more effective surveillance. No details are known about their mating systems or orientation towards host plants. Most probably, visual and olfactory elements play an important role and could be essential for the improvement of more sophisticated, highly effective, and selective monitoring methods. As far as we know, visual orientation and communication have greater relative importance compared to olfaction in Buprestidae [38,39,40]. Within the buprestids, *A. planipennis* has the only identified sex pheromone. The practical application for monitoring purposes with the use of this compound is successful only when applied together with green color and a green leaf volatile, (*Z*)-3-hexenol [41,42,43,44]. To this end, pheromones are often useful for detecting conspecifics in sparse populations interspersed with other insect groups like moths or bark beetles.

## Figures and Tables

**Figure 1 insects-14-00961-f001:**
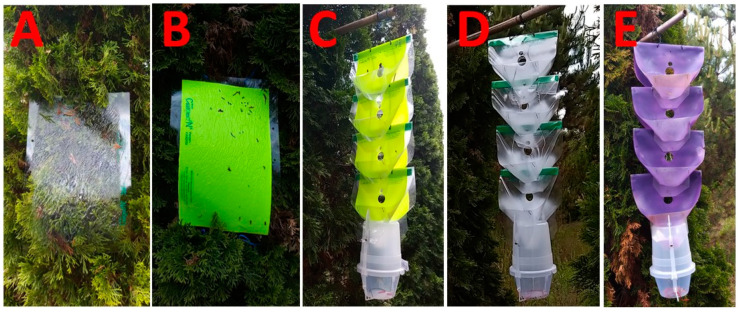
Photograph of the transparent sticky PAL (**A**), light green sticky PALz (**B**), light green non-sticky multifunnel MULTz (**C**), transparent non-sticky multifunnel MULT (**D**), and purple non-sticky multifunnel MULTp (**E**) traps.

**Figure 2 insects-14-00961-f002:**
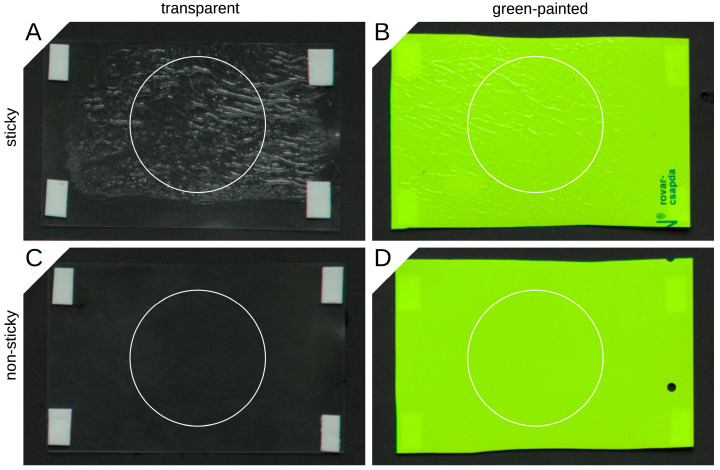
Photograph of the sticky ((**A**)—PAL) and non-sticky ((**C**)—MULT) transparent and green-painted surfaces ((**B**)—PALz; (**D**)—MULTz, respectively). White circles represent the areas to which the cosine corrector of the spectrometer was directed during the spectral measurements.

**Figure 3 insects-14-00961-f003:**
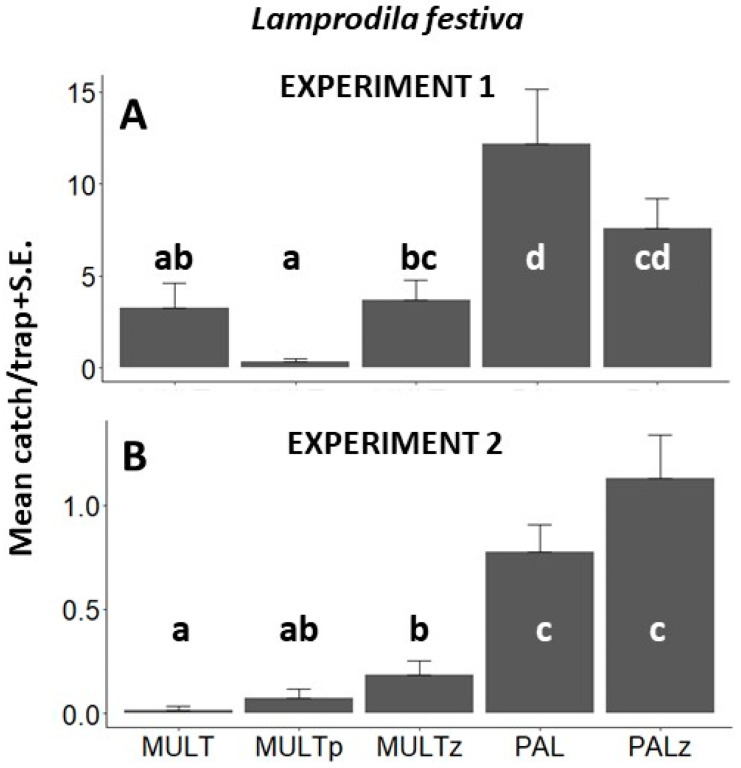
Mean (±S.E.) numbers of *Lamprodila festiva* specimens captured per trapping period in Exp. 1 ((**A**) Tahi, Hungary, 2019) and in Exp. 2 ((**B**) Tahi, Hungary, 2020) by transparent MULT, purple MULTp, and light green MULTz multifunnel traps and transparent PAL, and light green PALz sticky traps (all traps without olfactory bait). Means with the same letter within a diagram are not significantly different at *p* = 0.05 by Kruskal–Wallis, followed by Wilcoxon non-parametric tests.

**Figure 4 insects-14-00961-f004:**
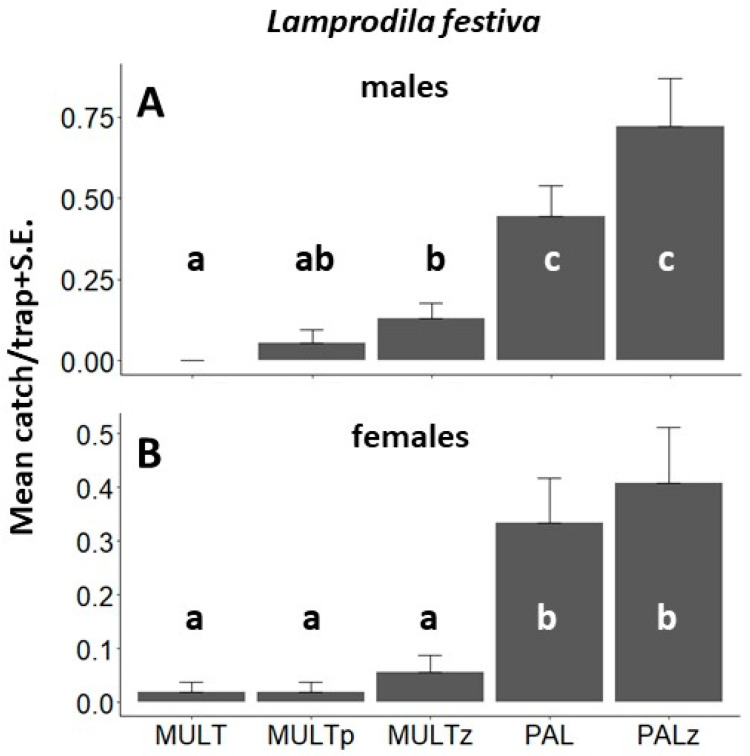
Mean (±S.E.) numbers of *Lamprodila festiva* specimens captured per trapping period in Exp. 2 (Tahi, Hungary, 2020) by transparent MULT, purple MULTp, and light green MULTz multifunnel traps and transparent PAL, and light green PALz sticky traps (all traps without olfactory bait). (**A**). males; (**B**). females. Means with the same letter within a diagram are not significantly different at *p* = 0.05 by Kruskal–Wallis, followed by Wilcoxon non-parametric tests.

**Figure 5 insects-14-00961-f005:**
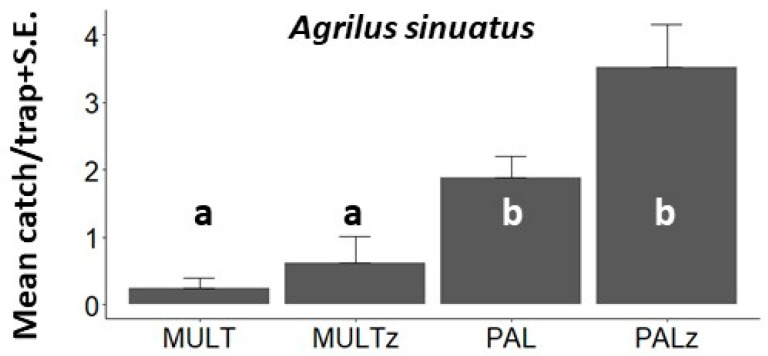
Mean (±S.E.) numbers of *Agrilus sinuatus* specimens captured per trapping period in Exp. 3 (Hajdúnánás, Hungary, 2019) by transparent MULT, light green MULTz multifunnel traps, transparent PAL, and light green PALz sticky cloak traps (all traps without olfactory bait). Means with the same letter within a diagram are not significantly different at *p* = 0.05 by Kruskal–Wallis, followed by Wilcoxon non-parametric tests.

**Figure 6 insects-14-00961-f006:**
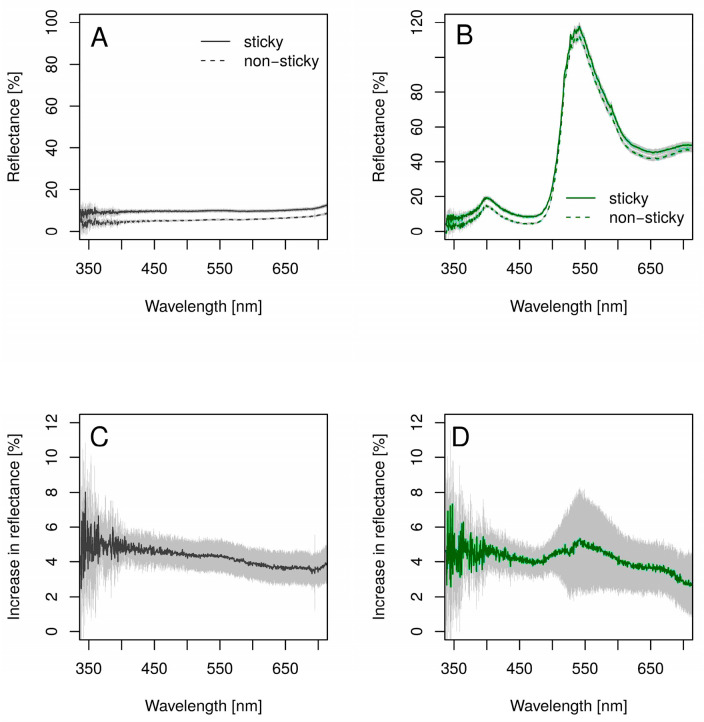
Reflectance spectra of the sticky and non-sticky surfaces on a black background and the increase in reflectance caused by the glittering/stickiness (**A**) Reflectance spectra of the transparent (sticky: PAL, non-sticky: MULT) surfaces. (**B**) Reflectance spectra of the green-painted (sticky: PALz, non-sticky: MULTz) surfaces. (**C**) Difference in reflectance spectrum of the sticky and non-sticky transparent surfaces (**D**) and green-painted surfaces. Each curve is the mean of 10 measurements. The light-colored area around each curve denotes the standard deviation.

## Data Availability

The data presented in this study are openly available at http://doi.org/10.6084/m9.figshare.24559795, accessed on 15 December 2023. (uploaded but not yet published).

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
