# Peer review of "The Optimal Choice of Trap Type for the Recently Spreading Jewel Beetle Pests Lamprodila festiva and Agrilus sinuatus (Coleoptera, Buprestidae)"

_insects, 2023, doi:10.3390/insects14120961_

Round 1

Reviewer 1 Report

Comments and Suggestions for Authors

This is a well written manuscript that describes several experiments conducted to determine the differences in catches of two species of buprestid beetles. As indicated below I think the description of the traps could be improved especially for the readers who are not familiar with these types of traps.  As such this paper does provide information about how to monitor for these beetles that are apparently increasingly becoming pests in Europe.

Materials and Methods – I find it difficult to visualize what the trap types look like. For readers who are not involved in trapping insects it would be helpful to supply figures of these traps.  In Figure 1 it would be helpful to label each trap type with the abbreviations used in the manuscript. It is unclear to me what the multifunnel trap looks like and where the colors are located on the trap.  I could not load the website for the source of the traps.

Line 267 – the non-target numbers should be reported.

Line 343 – it is unclear what the role of pheromones play in Buprestidae when olfaction is not utilized in mate finding.  Clarify or remove this sentence.

Comments on the Quality of English Language

Well written manuscript. Line 70 change form to from.

Author Response

Thank you for your efforts in reading our manuscript, forming an opinion, giving us helpful suggestions for the improvement of our work, and finally for your support in publishing the manuscript.

A figure (Figure 1) was added to help the understanding of the readers who are not familiar with these types of traps. We hope that the figure makes clear what the multi funnel trap looks like and where the colors are located on the trap.

In the former Figure 1 (now Figure 2) no trap types, but different type of surfaces was studied. This is the reason why we did not use the names of the trap types.

In the label of former Figure 1, now Figure 2 in the legends the reference letters are included to show which trap surface represents a specific trap type in the manuscript.

Indeed, we also experienced a problem in loading the website for the source of the traps, which has changed recently, and the website is available now.

Line 70 The change form to from was done.

Line 267 – the non-target numbers should be reported: Line 184-186 has been modified to indicate that only jewel beetles were recorded and identified to species. Most non-jewel beetle specimens were flies, which were not in focus in the present research. Their identification needs a significant effort, for which we had no experimental indication that it makes sense.

Line 343 – The unclear role pheromones play in Buprestidae was explained more in detail.

Reviewer 2 Report

Comments and Suggestions for Authors

This study reports on the comparison and contrast between the effectiveness of different trap types (colour and design) for monitoring jewel beetles, specifically Lamprodila festiva and Agrilus sinuatus.

It provide useful and important cost-effective recommendations for traps which could be developed for the detection and monitoring of these beetle for decision support in IPM programmes.

The manuscript is well written, clear and concise.

A few minor comments and questions should be considered.

Methods

Photographs of the traps would be a useful addition to the methods where you describe the traps.

Were the traps placed on a particular aspect of the trees in the plots? Was this considered?

Results

A comment on why more L. festiva were captured in 2019 compared to 2020 (perhaps in the discussion)

Discussion

Line 267 – non-target catches can be reduced using transparent sheets. There is no data in relation to the non-target catches in the results. This could be included, even if only to compare the numbers of other species captured between the traps and the proportion for the trap catch (target v non-target). This may be an important consideration is some situations when selecting a particular trap.

When comparing your data with other studies trapping buprestids some further comment on whether these studies utilised lures or not.

Author Response

Thank you for your efforts in reading our manuscript, forming an opinion, giving us helpful suggestions for the improvement of our work, and finally for your support in publishing the manuscript.

A figure (Figure 1) was added to help the understanding for the readers who are not familiar with these types of traps.

Reviewer 2: Were the traps placed on a particular aspect of the trees in the plots? Was this considered?

Answer: Yes, traps were placed on the sunlit sides of trees, which is now indicated in the modified text in lines 173  and 178.

Reviewer 2: Results. A comment on why more L. festiva were captured in 2019 compared to 2020 (perhaps in the discussion):

Answer: Line 341-345 The possible reason for the decrease in catches was described.

Reviewer 2: Discussion. Line 267 – non-target catches can be reduced using transparent sheets. There is no data in relation to the non-target catches in the results. This could be included, even if only to compare the numbers of other species captured between the traps and the proportion for the trap catch (target v non-target). This may be an important consideration is some situations when selecting a particular trap.

Answer: Line 184-186 has been modified to indicate that only jewel beetles were recorded and identified to species. Most non-jewel beetle specimens were flies, which were not in focus in the present research. Their identification needs a significant effort, for which we had no experimental indication that it makes sense.

Reviewer 2: When comparing your data with other studies trapping buprestids some further comment on whether these studies utilised lures or not.

Answer: Line 118 states that efficient tools for the detection or monitoring of L. festiva and A. sinuatus are lacking to date. In the material and methods, olfactory baits are not mentioned.

Line 379 of the latest version: Within the buprestids, A. planipennis has the only identified sex pheromone, and the practical application for monitoring of this compound is successful only when applied together with green color and a green leaf volatile, (Z)-3-hexenol [41-44].  

Although it is not stated in the manuscript, there are no other olfaction-based practical applications for Buprestids in practice.

Reviewer 3 Report

Comments and Suggestions for Authors

See attached file "Comments"

One more thing: The table with the P-values of KW and MW, whi not put the statistics of the tests in that table too (an not only the p-values)? In that way you could remove all the redious X2 and p-values from the text...and reduce this redundancy

Author Response

Thank you for your efforts in reading our manuscript, forming an opinion, giving us helpful suggestions for the improvement of our work, and finally for your support in publishing the manuscript.

Line 212-213, 233, 237, 247-248 χ2 (df) was removed from text and inserted in Table A1 3rd column.

Reviewer 3: 1-L100-101. " As possible attractive visual cues, light green (sometimes called fluorescent yellow) and purple colours were tested" Why? Why only these colors. Please provide a little bit more of additional background.

Answer: Line 121-126 - Citations were added which show that green color is attractive to several Agrilus spp., while purple is somewhat less attractive to fewer species, but is still considered a possible attractive colour.

Reviewer 3: Methods - The terminology used to name the treatments (which follow brand names of the traps) makes it very difficult for the unfamiliar to follow up which treatment is which. It would have been a lot better to use letters…

Answer: The terminology used to name the treatments, which follow code names of the traps are not brands. These code names are often used in publications. It might be another good approach to use letters but needs a complete rewriting of the manuscript and all figures, without giving that much to the understanding of the paper.

Reviewer 3: 3-L162-163. Since the light source is natural light, and since the chromatic composition of natural light varies with sun´s azimuth, please indicate the geographical location, day of the year and time of the day when the light measurements were made.

Answer: The measurement location and time were added to this paragraph as follows:

A total number of 10 measurements were made on each surface (location: N 47° 28’ 43.586”, E 19° 1’ 51.99”, time: 28 July 2023, 12:20 UTC + 2h).

Reviewer 3: 4-L164 "and a Spectralon was used". Should define what type of device "Spectralon" is. Spectralon is the commercial name of a product from Ocean Optics brand. But what is exactily Spectralon? What does it do?

Answer: In the revised materials and methods we wrote the following:

...with a P400-010-UV-VIS fiber (Ocean Optics, Largo, USA) and a Spectralon® white diffuse reflectance standard (Edmund Optics Inc., Barrington, USA) was used for the measurements. Spectralon® is a diffuse reflector possessing a constant 99% reflectance over a wide spectral range including the region of insect vision.

Reviewer 3: 5-L165. Field of view 20 degrees. But what was the distance between the fiber optic and the reflecting surface?

Answer: The distance between the fiber optic and the reflecting surface was already included in the manuscript at line 168: “(distance = 5 cm)”.

To make it more clear, we added the following sentence:

The distance between the fiber terminal and the reflecting surface was 5 cm.

Reviewer 3: 6-L171-172. Please cite the authors of the R libraries

Answer: We checked and found that the following authors are mentioned in the references:

  1. R Core Team. R: a language and environment for statistical computing. R Foundation for Statistical Computing, Vienna, Austria. https://www.R-project.org/. 2022.
  2. Wickham, H.; Francois, R.; Henry, L.; Müller, K. dplyr: A grammar of data manipulation. R package version 0.7.4. https://CRAN.R-project.org/package=dplyr. 2017.
  3. Wickham, H. ggplot2: Elegant graphics for data analysis; Springer-Verlag: New York, 2009.

Reviewer 3: Statistical analysis

Answer: Thank you for your suggestions regarding the more up-to-date statistical analysis. In the present study, we cannot see the point of redoing the analysis and rewriting the manuscript, although we agree with your approach.